# Transmural Ultrasonography in the Evaluation of Horse Hoof Internal Structures: Comparative Quantitative Findings—Part 2

**DOI:** 10.3390/ani13142267

**Published:** 2023-07-11

**Authors:** Andrés Felipe Castro-Mesa, Rafael Resende Faleiros, José Ramón Martínez-Aranzales

**Affiliations:** 1Equine Medicine and Surgery Research Line (LIMCE), CENTAURO Research Group, School of Veterinary Medicine, Faculty of Agricultural Sciences, Universidad de Antioquia, Medellín 050010, Colombia; jose.martinez@udea.edu.co; 2Equinova Research Group, School of Veterinary, Universidad Federal de Minas Gerais, Belo Horizonte 31270-901, Brazil; faleirosufmg@gmail.com

**Keywords:** transmural ultrasound, hoof, distal phalanx, lamellar layer, anatomical relationship

## Abstract

**Simple Summary:**

The internal stratum (lamellar) of the equine hoof is composed of the interdigitated dermal and epidermal lamellae, forming the dermo-epidermal junction (DEJ). In cases of laminitis, the configuration of the lamellae can be affected, leading to changes in the position of the distal phalanx in chronic cases. Digital radiography (DR) is a routine diagnostic imaging technique used to detect, measure, and monitor such changes. The objective of the study was to compare the use of the transmural ultrasound technique and digital radiography (DR) to detect the hoof strata and structures within, as well as to measure the spatial relationship between the distal phalanx and the hoof capsule. This ultrasonographic transmural technique enables the determination and measurement of the ultrasonographic distal phalanx–hoof relationship.

**Abstract:**

The transmural ultrasound allows for the definition of the normal appearance of the hoof tissues and internal structures. Determining such measurements and proportions could contribute to evaluations of the normal spatial distal phalanx–hoof relationship. The objective was to describe the hoof’s dermis and epidermis measurements using the transmural technique, in comparison with DR and anatomical sections. Sixty-two healthy hooves without digital radiographic abnormalities, made up of 30 anatomical pieces (phase 1) and 32 alive horse hooves (phase 2), and 16 sagittal sections of the first ones were used. The proximal and distal planes were compared, defining the following measurements: coronary band-extensor process, distal phalanx apex–hoof wall, sole dermis, middle stratum, parietal dermis, lamellar layer, the sublamellar dermis, and ratios between them. Most of the measurements were consistent among methods. Some showed measurement differences and a minority was impossible to determine. The transmural technique allowed for the observation and replication of measurements of the hoof structures, described with other diagnostic aids. It also allowed for the implementation of new measurements that would help to determine the spatial location of the distal phalanx. Additionally, it contributes to the visualization of normal findings, which will be useful to compare variations in the different phases of laminitis.

## 1. Introduction

The transmural ultrasound technique (a sagittal axis evaluation) was able to describe the normal appearance of the medium layer (tubular wall), the dermo-epidermal junction (DEJ) and the sublamellar dermis of the hoof in the parietal region [1]. The parietal dermo-epidermal junction (DEJ), also known as the internal (lamellar) layer of the wall, is formed by the interdigitation of the dermal and epidermal layers of the hoof. This layer makes up the suspensory apparatus of the distal phalanx (SADP), which aims to stabilize and protect the digit from load forces [2,3,4]. Regardless of its origin, injury to this layer is a common occurrence in cases of laminitis, often resulting in the collapse of the distal phalanx [5,6,7,8,9,10]. The sublamellar dermis is the other component of the parietal dermis region and consists of blood vessels and collagen fibers that connect the dermal lamellae to the foramina on the parietal surface of the distal phalanx [2,3,4].

Digital radiography has been used to describe the normal position of the distal phalanx [11,12,13,14,15,16,17] and the layers of the hoof, but without differentiating the components of the parietal dermis [18,19]. However, advanced imaging diagnostic techniques such as magnetic resonance imaging (MRI) have made it possible to differentiate between the tubular wall and lamellar layer, as well as the lamellar layer and sublamellar dermis [14,19,20].

More recently, transmural ultrasound has been used to distinguish between all these layers, highlighting the importance of this diagnostic imaging modality [1]. While the keratinized tissue of the hoof wall was previously a limitation to ultrasound penetration [21,22,23], submersion of the hooves has improved this issue. The transmural ultrasound technique also allowed for the description of other structures, such as the coronary papillae, extensor process, parietal surface, and apex of the distal phalanx, as well as the papillae and dermis of the sole. These structures serve as reference points to locate the distal phalanx within the hoof [1].

Determining the anatomical relationship between the distal phalanx and the hoof capsule is diagnostically valuable, and radiographic evaluation is routinely used to estimate distances, angulations, and proportions to determine normality and deviations [11,12,13,14,15,16,17]. However, ultrasonography has not traditionally been considered a routine diagnostic aid for this purpose [21,22,23,24,25,26,27,28,29,30,31,32]. Previous studies using digital radiology and MRI have described the differences between the wall-soft tissues and the lamellar layer–sublamellar dermis, respectively [19]. However, transmural ultrasonography allows for the diagnostic imaging of all these structures, indicating its relevance and pertinence as a diagnostic tool. The findings of this study may have prognostic utility in cases that compromise these structures, such as laminitis [13].

Therefore, the objective of this study is to determine the diagnostic value of transmural ultrasonography in comparison to digital radiography and anatomical sections in estimations of normal measurements and ratios of the hoof’s dermis and epidermis, as well as the ratios of the lamellar layer and the sublamellar dermis with reference to the parietal dermis. The study aims to estimate the spatial relationship of the soft tissues and the distal phalanx, which could be useful in identifying and managing the conditions that affect these structures.

## 2. Materials and Methods

### 2.1. Anatomical Pieces and Alive Horse Hooves

This research was approved by the Ethics Committee for Animal Experimentation of the Universidad de Antioquia, Colombia (Act number 135, 2 September 2020). A total of 62 healthy hooves of thoracic limbs (TL) and pelvic limbs (PL) from Colombian creole horses were distributed in two study phases. In phase 1, 30 hooves collected in a slaughterhouse were disarticulated from the fetlock joint, transported fresh for 4 h, and finally frozen at −21 °C for 150 days. In phase 2, 32 hooves of eight healthy mares without lameness and with an average age, weight, and body condition score [33] of 6.9 ± 2.97 years, 284.9 ± 32.1 kg, and 4.1 ± 0.44 (respectively) were selected based on their normal condition at clinic inspection. 

The hooves were selected based on a negative lameness inspection by assessing the lameness at walk and light trot using hoof testers, and percussion was also carried out in all horses, including on horses that were the donors of the pieces. 

All hooves were also evaluated using digital radiography to rule out any alterations in internal and external structures. Lateral–medial projections were taken using a portable veterinary X-ray equipment, generator MINXRAY^®^ HFX90V (MinXray, Inc., Northbrook, IL, USA), with 1.2 mAs—66 kv settings. The location and normal anatomy of the distal phalanx were determined by measuring angles and other relevant factors (radiological software Accuvet^®^ version 1.1.0 developed by RadmediX), with relevance to laminitis [11,12,13,14,15,16,17].

### 2.2. Measurements of the Hoof Internal Structures and Ultrasonographic Ratios

The measurements evaluated in the sagittal plane sections of the anatomical pieces (Figure 1), digital radiographs (Figure 2), and ultrasonography (Figure 3) were the tubular hoof wall (P), the parietal dermis (D), and its components the lamellar layer (L) and sublamellar dermis (C). These assessments were made on two levels that were perpendicular to the parietal surface.

With a ultrasonographic sagittal view, the distal interphalangeal joint (DIPJ) and the extensor process [29] were identified and distally explored, and the change in inclination between the extensor process and the parietal surface at the proximal level was determined [18]. The distal level was located approximately 6 mm proximal to the apex of the distal phalanx [19]. 

The coronary band-extensor process (A), the distal phalanx apex–hoof wall (B), and the sole dermis (S) distances were also determined in a ultrasonographic sagittal view, with the exception of distance A, which was determined using the transverse view [1] at the level of the coronary band [29]. Distance A was determined by a diagonal line, and distances B and S were determinate by a perpendicular line to the parietal surface at the level of the apex of the distal phalanx. This line was prolonged to the hoof wall and to the sole papillae, determining the respective measurements. 

In the latero-medial radiographs, the contrast-adjusted technique permitted the differentiation of the soft tissue from the others with a radiolucent and radiodense appearances, respectively [14,18,19]. However, the components of the parietal dermis, the internal (lamellar) layer and the sublamellar dermis [2,3,4] could not be differentiated by digital radiography [14,18,19]. 

To obtain ultrasonographic measurements, all hooves were prepared and submerged in water or ice and water in order to differentiate the middle layer (tubular), the internal layer (lamellar), and the sublamellar dermis, as well as the transitions between them. This methodology and the findings were described in a companion article titled Transmural Ultrasonography in the Evaluation of Horse Hoof Internal Structures: Comparative Qualitative Findings—Part 1 [1].

The sagittal plane sections were generated from 16 hooves in anatomical pieces (phase 1) to permit replication of the previously mentioned measurements and evaluate the parietal dermis and its components, which were also obtained with the transmural ultrasound.

In addition, ultrasonographic ratios were evaluated with the aim of reinforcing the identification of the normal position of the distal phalanx inside the hoof. The evaluated ratios were the parietal dermis: coronary band-extensor process (D1:A and D2:A); the lamellar layer:parietal dermis (L:D) and the sublamellar dermis:parietal dermis (C:D) at the proximal and distal levels; the apex–hoof wall:coronary band-extensor process (B:A) and the sole dermis:coronary band-extensor process (S:A).

### 2.3. Statistical Analysis

Descriptive statistics were first calculated as measures of central tendency (i.e., interquartile range (IQR), mean (average), median) and measures of dispersion (i.e., standard deviation (SD), coefficient of variation (CV)). Non-normally distributed data were analyzed using the median. In phase 1, we compared the distances A, D1, D2, B, and S, obtained from transmural ultrasonography, digital radiography, and anatomical sections using the nonparametric Friedman test. We also compared the measurements of the parietal dermis and its components (the lamellar layer and the sublamellar dermis) obtained from transmural ultrasonography and anatomical sections using the nonparametric Wilcoxon test. In phase 2, we compared the distance measurements of A, D1, D2, B, and S obtained from transmural ultrasonography and digital radiography using the Wilcoxon test.

All data were analyzed using STATA^®^, version 16.1 (StataCorp., College Station, TX, USA) considering a statistical significance level of *p* ≤ 0.05.

## 3. Results

The measurements obtained from digital radiography (DR), transmural ultrasound (US), and sagittal sections (AN) in phase 1 are presented in Table 1 and Table 2 for the TL and PL, respectively. Numerically, the radiographic measurements were greater, and the ultrasonographic measurements were smaller when compared to the three methods, except for the parietal dermis (D), which was greater in the ultrasound measurements.

In Table 1 for the TL, the parietal dermis at the proximal and distal levels (D1 and D2), the distal phalanx apex–hoof wall (B), and the sole dermis (S) were similar in the DR, US, and AN evaluations. The sublamellar dermis at the distal level (C2) was similar in the US and AN evaluations. In the ultrasonographic ratio evaluation, the approximate thicknesses of the parietal dermis at the proximal and distal levels with reference to the coronary band-extensor process (D:A) were 70% and 66%, respectively. The lamellar layers with reference to the parietal dermis (L:D) were 71% and 68% in the proximal and distal levels, respectively. The sublamellar dermises with reference to the parietal dermis (C:D) were 28% and 30% in the proximal and distal levels, respectively. The distal phalanx apex–hoof wall with reference to the coronary band-extensor process (B:A) was 84%, and the sole dermis with reference to the coronary band-extensor process (S:A) was 44%.

In Table 2 for the PL, the parietal dermis at the proximal and distal levels (D1 and D2) and the sole dermis (S) were similar in the DR, US, and AN evaluations. The parietal dermis at the proximal level (D1) and the sublamellar dermis at the proximal and distal levels (C1 and C2) were similar in the US and AN evaluations. In the ultrasonographic ratio evaluation, the approximate thickness of the parietal dermis at the proximal and distal levels with reference to the coronary band-extensor process (D:A) was 71% and 68%, respectively. The lamellar layer with reference to the parietal dermis (L:D) was 74% and 69% in the proximal and distal levels, respectively. The sublamellar dermis with reference to the parietal dermis (C:D) was 29% at both levels. The distal phalanx apex–hoof wall with reference to the coronary band-extensor process (B:A) was 74%, and the sole dermis with reference to the coronary band-extensor process (S:A) was 36%.

Table 3 and Table 4 present the measurements obtained in phase 2 for TL and PL, respectively, using digital radiography (DR) and transmural ultrasound (US). The radiographic measurements were generally greater, except for the parietal dermis (D), which was greater in the ultrasonographic evaluation. In both types of limbs, the distances of the distal phalanx apex–hoof wall were similar in the DR and US evaluations.

In the ultrasonographic ratios evaluation of the TL (Table 3), the approximate thicknesses of the parietal dermis at the proximal and distal levels with reference to the coronary band-extensor process (D:A) were 73% and 70%, respectively; the lamellar layers with reference to the parietal dermis (L:D) were 67% in both the proximal and distal levels; the sublamellar dermises with reference to the parietal dermis (C:D) were 34% and 33% at the proximal and distal levels, respectively. The distal phalanx apex–hoof wall with reference to the coronary band-extensor process (B:A) was 70%, and the sole dermis with reference to the coronary band-extensor process (S:A) was 32%.

In the ultrasonographic ratios evaluation of the PL (Table 4), the approximate thicknesses of the parietal dermis at the proximal and distal levels with reference to the coronary band-extensor process (D:A) were 73% and 69%, respectively; the lamellar layers with reference to the parietal dermis (L:D) were 67% at both the proximal and distal levels; the sublamellar dermises with reference to the parietal dermis (C:D) were 32% at both levels. The distal phalanx apex–hoof wall with reference to the coronary band-extensor process (B:A) was 83%, and the sole dermis with reference to the coronary band-extensor process (S:A) was 30%.

## 4. Discussion

Ultrasonography is a widely used technique for the diagnosis of lameness in horses [32]. Several approaches have been described in the literature to assess some structures within the hoof [21,22,23,24,25,26,27,28,29,30,31,32]. However, a recent study described a new ultrasonographic approach, the transmural technique, after hydrating the hoof wall through water immersion [1]. This technique enabled visualization of the extensor process, parietal surface, and apex of the distal phalanx; coronary papillae, middle layer (tubular wall), and transition to the inner layer (lamellar); parietal dermis, where the transition from sublamellar dermis to lamellar dermis occurs, defining the lamellar layer between the transition zones; and papillae and sole dermis [1]. Before that, the differentiation of layers that comprise the parietal dermis was only reported by MRI [14,19,20,34]. 

In this study, the transmural ultrasonographic technique was compared with radiology and anatomical sections, obtaining a qualitative description of the previously mentioned structures. However, in addition to the ultrasonographic anatomy, it is important to define the distances between the internal structures of the digit to establish the spatial relationship or location of the third phalanx with relevance to the adjacent structures. Therefore, this study aimed to obtain quantitative information based on measurements and ratios in anatomical pieces and alive horse hooves.

The obtention of larger radiographic measurements when compared with macroscopic anatomical measurements (millimeter ruler) in anatomical sections and MRI has also been reported [18,19]. The anatomical pieces were frozen, as formerly reported [19]. This condition may have limited the radiographs’ ability to differentiate the keratinized tissues and non-keratinized tissues. However, this was not an issue when using ultrasonography, as the hydration of the hoof wall likely allowed for the differentiation of the keratinized tissues.

When comparing the measurements of the parietal dermis (D) and its layers (L and C) between the ultrasonography and the anatomical sections, several factors could have influenced the results. For instance, the anatomical pieces were not individually packed in plastic bags to prevent desiccation [19] and underwent freezing and thawing processes for the development of other procedures, which could cause tissue damage [35,36]. In addition, if the anatomical sections in the sagittal plane are not made millimetrically, this can generate distortion factors and alter the configuration of the limits between the layers [18].

While the stereomicroscope or distance of the camera lens were not effective in determining configuration changes in the tissues and the limits between L-C and D-P, which were described in the qualitative findings of the transmural technique [1], histological sections could improve these aspects and confirm alterations to or normality of the hooves’ soft tissues [19,20,37,38,39]. The implementation of in vivo lamellar biopsy procedures with sedation techniques and regional anesthesia has also been described [38,39]. However, the invasive nature of the procedure could limit its applicability. In future studies, the use of histological sections could provide more accurate information about the spatial relationship and configuration of the internal structures of the hoof.

These observations are supported by the fact that the distance D was not statistically similar between ultrasonography and anatomical sections in Table 1 and Table 2. Therefore, the implementation of histological sections in future studies would improve these aspects and confirm alterations to or the normality of the hooves´ soft tissues [19,20,37,38,39]. However, the invasive nature of the in vivo lamellar biopsy procedure [38,39] could limit its applicability. The measurement of distance B, involving the tubular hoof wall (P), showed consistency between the methods used. However, there are several factors that can influence the keratinize tissue’s thickness, such as breed and size, the presence of horseshoe, environmental conditions, poor conformation, and poor hoof trimming [16,19,40,41]. Additionally, the variation in the phalangeal axis is associated with flexural deformities of the distal interphalangeal joint [14].

Distances A, B, and S were previously described in radiography and can also be measured by ultrasonography. These measurements are of great importance in establishing the positioning of the third phalanx in conditions that affect its support, such as chronic laminitis [14,42,43,44]. However, there are some limitations to using ultrasonography to measure these distances, such as the refraction phenomenon, which can cause distortions in the image and the displacement of the anatomy from its original position [45]. In this study, the measurement of distance S in ultrasonography was found to be lower than that in previous radiographic studies, which may be attributed to this phenomenon. Nevertheless, the measurements of A, B, and S in digital radiography, ultrasonography, and anatomical sections analyzed herein showed a similar pattern.

The measurements made towards the distal level of the parietal surface of the distal phalanx are also smaller when compared to previous radiographic studies [41]. A similar pattern was found in the measurements of digital radiography, ultrasonography, and anatomical sections analyzed herein.

The transmural technique used in this study required different approaches for measuring distances A, B, and S. Distance A was determined using a transverse view at the level of the coronary band [29] as a diagonal line, while distances B and S were measured using a sagittal ultrasonographic view at the level of the apex of the distal phalanx with a perpendicular line to the parietal surface. This differs from previous studies that used radiography and MRI, where the distances were determined using a perpendicular line to the ground [13,14,17,19,44]. 

However, due to the interaction between ultrasound and the distal phalanx, it is impossible to detect its inclination and to estimate the ratios associated with palmar cortical length [11,12,13,14,15,16,17,19,44,46]. To overcome this limitation, the ratios between distances B:A, S:A, and, D:A, L:D and C:D at the proximal and distal levels were proposed to determine the spatial relationship of soft tissues and the distal phalanx. The use of ratios has been reported in previous studies to determine normality in thoracic limbs and compare dorsal and sole dermis thickness with palmar cortical length in digital radiography and the MRI of anatomical specimens [18,19].

The ratios of the parietal dermis and the layers that compose it (L:D and C:D) would increase in cases of symmetrical distal displacement or “sinking” [14,40,41]. In cases of capsular rotation or phalangeal rotation [14,44], the layers would be greater in the distal. In addition, a simultaneous decrease in B:A and S:A, or the absence of the former, would indicate a severe process or that a longer immersion time is necessary for diagnosis [1]. A combination of these results in the case of simultaneous dorsal and distal displacement. Regarding D:A ratio, it would increase in cases of rotation but can be maintained in cases of collapse. Therefore, each measurement must be interpreted individually and correlated with the cited ratios and the qualitative findings of the hoof layers [1].

The information provided here may be a starting point for the diagnosis of chronic laminitis by ultrasonography. Although it exceeded the stated objectives, incidental findings [47] were reported during the search for horses to be used in the present study. As seen in Figure A1, these changes included classical signs of chronic laminitis, such as an increase in distance L and a loss of the parallel alignment between the parietal dermis and the periosteum (modified fibrocartilage) of the distal phalanx [2,3,4,48].

The changes in the positioning of the distal phalanx were better explained in the anatomical pieces of a horse subjected to euthanasia due to chronic laminitis (Figure A2). The increase in A and B, the progressive distal increase in D, L, and C, and the absence of S (Figure A2b) demonstrated symmetrical distal displacement and capsular rotation, as previously detected by radiology. Various findings have been associated with chronic laminitis. These include “sinking” [14,41,43,44], the collapse of coronary veins and arteries, alterations in the coronary band, as observed with MRI [49], sole dermal necrosis [8], vascular changes at the sublamellar level, as seen in digital venography [50,51,52], and the appearance of newly formed tissue referred to as the lamellar wedge [53].

The ultrasonographic measurements of the spaces, layers, and spatial relationship of the internal structures of the hoof, as determined in this study, can provide several advantages in similar cases of chronic laminitis. By using this technique, clinicians can obtain detailed and accurate information about the condition of the hoof, which can aid in diagnosis and treatment planning. Therefore, ultrasonography can be considered a valuable tool in the management of chronic laminitis. To further validate the usefulness of these measurements, it is recommended that future studies be conducted using MRI and histological sections of normal horses, as well as natural or induced clinical cases of laminitis. This will help to objectively determine the suitability of these measures for clinical use and improve our understanding of chronic laminitis.

## 5. Conclusions

The transmural ultrasonographic technique used in this study allowed for the observation and replication of measurements of the internal structures of the hoof that have been described by other diagnostic aids. Additionally, this technique provided new measurements that can aid in determining the spatial location of the distal phalanx within the hoof.

## Figures and Tables

**Figure 1 animals-13-02267-f001:**
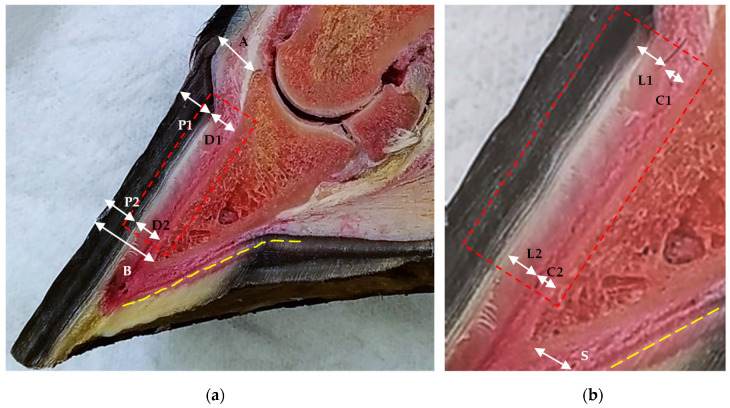
Anatomical sagittal plane section of an equine thoracic hoof. (**a**) Note the coronary band-extensor process distance (A), the tubular hoof wall (P1 and P2) thickness, and the parietal dermis (D1 and D2) thickness at the proximal and distal levels, respectively. Additionally, note the distal phalanx apex–hoof wall distance (B) and the sole papillae (yellow broken line). (**b**) Enlarged detail. Proximal is to the right. Note the lamellar layer (L1 and L2) and the sublamellar dermis (C1 and C2) thickness at the proximal and distal levels, respectively. Sole dermis (S). Sole papillae (yellow broken line). (Saw v25-19, 220v/60/2f-1hp/JAVAR^®^; digital stereomicroscope Olympus^®^ SZX7, camera DP27, manufactured by Evident Corporation, Inatomi, Tatsuno-machi, Kamiina-gun, Nagano, Japan, and Olympus cellSens Standard software version 2.3.)

**Figure 2 animals-13-02267-f002:**
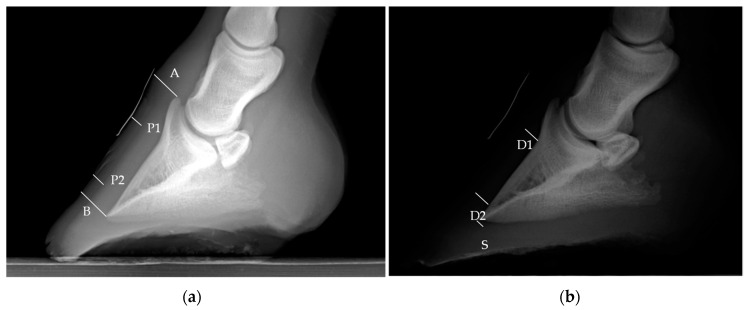
Latero-medial digital hoof radiographs of an equine thoracic limb, where the contrast adjustment allows to differentiate the soft tissues from the keratinized ones. (**a**) Note the coronary band-extensor process (A) and distal phalanx apex–hoof wall (B), the tubular hoof wall distances (P1 and P2) at the proximal and distal levels, respectively. (**b**) Deep radiographic layer or parietal dermis (D1 and D2) at the proximal and distal levels, respectively. Sole dermis (S). (1.2 mAs—66 kv/generator MINXRAY^®^ HFX90V, radiological software Accuvet^®^ version 1.1.0 developed by RadmediX.)

**Figure 3 animals-13-02267-f003:**
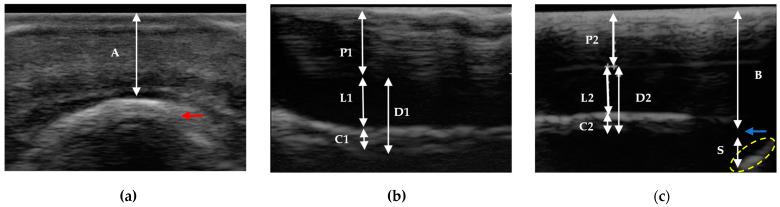
Transmural ultrasonographic technique of an equine hoof. (**a**) Transverse evaluation on the coronary band of a thoracic limb. Note the extensor process (red arrow) and the coronary band-extensor process distance (A). (**b**,**c**) Sagittal evaluation of a pelvic limb. Proximal is to the left. Note the tubular hoof wall (P1 and P2), the parietal dermis (D1 and D2), the lamellar layer (L1 and L2), and the sublamellar dermis (C1 and C2) thickness, at the proximal and distal levels, respectively. Additionally, note the apex of the distal phalanx (blue arrow), the distal phalanx apex–hoof wall distance (B), the sole papillae (yellow oval), and the sole dermis (S). (Ultrasound machine G30 Color Doppler—Emperor Medical, EMP^®^ Shenzhen Emperor Electronic Technology Co., Ltd., Nanshan district, Shenzhen, China; lineal transducer for tendons 6.5 mHz.)

**Table 1 animals-13-02267-t001:** Measurements (mm) obtained from thoracic limbs hooves in anatomical pieces through digital radiography, ultrasonography, and anatomical sections of the anatomical pieces.

Variables	DR	US	AN	*n* = 15	
	Median ± SD	Median ± SD	Median ± SD	*p* Value	
Distance A	14.1 ± 1.61	10.8 ± 1.51	13.67 ± 1.57	0.02	-
Tubular hoof wall (P1)	9.0 ± 1.15	6.6 ± 1.11	8.79 ± 1.13	-	-
Parietal dermis (D1)	7.1 ± 0.30 ^a^	8.1 ± 0.96 ^a^	5.7 ± 0.37 ^a^	0.19	0.01
Lamellar layer (L1)	-	5.8 ± 1.13	2.5 ± 0.68	-	0.01
Sublamellar dermis (C1)	-	2.25 ± 0.36	3.1 ± 0.52	-	0.01
Tubular hoof wall (P2)	8.7 ± 0.95	6.4 ± 0.64	8.58 ± 0.77	-	-
Parietal dermis (D2)	6.2 ± 0.62 ^a^	7.3 ± 0.60 ^a^	4.62 ± 0.67 ^a^	0.07	0.02
Lamellar layer (L2)	-	5.0 ± 0.97	2.3 ± 0.53	-	0.01
Sublamellar dermis (C2)	-	2.1 ± 0.48 ^b^	2.37 ± 0.46 ^b^	-	0.67
Distance B	15.2 ± 1.22 ^a^	13.8 ± 0.73 ^a^	14.1 ± 1.58 ^a^	0.07	-
Sole dermis (S)	6.0 ± 1.31 ^a^	4.9 ± 1.32 ^a^	6.17 ± 0.65 ^a^	0.12	-
**US ratios**	**Proximal**		**Distal**		
	Mean ± SD		Mean ± SD		
D: A	**0.70 ± 0.03**		**0.66 ± 0.03**		
L: D	0.71 ± 0.02		0.68 ± 0.02		
C: D	0.28 ± 0.01		0.30 ± 0.02		
	Mean ± SD				
B: A	0.84 ± 0.05				
S: A	0.44 ± 0.02				

DR: Digital radiology. US: ultrasonography. AN: anatomical sections. SD: standard deviation. ^a,b^ Columns with different letters indicates non-statistical significance (*p* ≥ 0.05) using the Friedman and Wilcoxon tests, respectively. Bold ratios were obtained of at least one of these variables.

**Table 2 animals-13-02267-t002:** Measurements (mm) obtained from pelvic limbs hooves in anatomical pieces through digital radiography, ultrasonography, and anatomical sections of the anatomical pieces.

Variables	DR	US	AN	*n* = 15	
	Median ± SD	Median ± SD	Median ± SD	*p* Value	
Distance A	14.7 ± 1.32	10.5 ± 1.04	14.08 ± 1.63	0.02	-
Tubular hoof wall (P1)	9.2 ± 1.23	6.6 ± 0.65	8.96 ± 1.11	-	-
Parietal dermis (D1)	6.4 ± 0.70 ^a^	7.85 ± 1.33 ^a,b^	5.3 ± 1.77 ^a,b^	0.43	0.25
Lamellar layer (L1)	-	5.6 ± 0.76	2.4 ± 0.35	-	0.01
Sublamellar dermis (C1) ^b^	-	2.1 ± 0.51 ^b^	2.7 ± 0.38 ^b^	-	0.16
Tubular hoof wall (P2)	8.9 ± 0.83	6.65 ± 0.72	8.3 ± 0.42	-	-
Parietal dermis (D2) ^a^	6.0 ± 0.90 ^a^	7.3 ± 0.58 ^a^	4.57 ± 0.59 ^a^	0.57	0.02
Lamellar layer (L2)	-	5.0 ± 1.23	1.96 ± 0.48	-	0.01
Sublamellar dermis (C2) ^b^	-	2.15 ± 0.94 ^b^	2.31 ± 0.38 ^b^	-	0.58
Distance B	16.55 ± 1.36	14.5 ± 0.81	14.56 ± 1.41	0.02	-
Sole dermis (S) ^a^	6.95 ± 1.0 ^a^	3.6 ± 0.87 ^a^	5.89 ± 0.88 ^a^	0.08	-
**US ratios**	**Proximal**		**Distal**		
	Mean ± SD		Mean ± SD		
D: A	**0.71 ± 0.04**		**0.68 ± 0.02**		
L: D	0.74 ± 0.02		0.69 ± 0.01		
C: D	0.29 ± 0.02		0.29 ± 0.01		
	Mean ± SD				
B: A	0.74 ± 0.02				
S: A	0.36 ± 0.03				

DR: digital radiology. US: ultrasonography. AN: anatomical sections. SD: standard deviation. ^a,b^ Columns with different letters indicates non-statistical significance (*p* ≥ 0.05) using the Friedman and Wilcoxon tests, respectively. Bold ratios were obtained of at least one of these variables.

**Table 3 animals-13-02267-t003:** Measurements (mm) obtained from thoracic limbs from the hooves of alive horses through digital radiography and ultrasonography.

Variables	DR	US		*n* = 16
	Median ± SD	Median ± SD		*p* Value
Distance A	13.3 ± 2.18	10.3 ± 1.24		0.001
Tubular hoof wall (P1)	8.5 ± 1.56	6.95 ± 0.55		-
Parietal dermis (D1)	6.3 ± 1.04	7.8 ± 0.58		0.001
Lamellar layer (L1)	-	5.05 ± 0.56		-
Sublamellar dermis (C1)	-	2.6 ± 0.25		-
Tubular hoof wall (P2)	7.55 ± 1.22	6.55 ± 0.8		-
Parietal dermis (D2)	6.7 ± 0.87	7.35 ± 0.58		0.019
Lamellar layer (L2)	-	5.15 ± 0.5		-
Sublamellar dermis (C2)	-	2.5 ± 0.23		-
Distance B	14.4 ± 1.06 ^a^	14.65 ± 0.72 ^a^		0.906
Sole dermis (S)	7.7 ± 1.58	3.3 ± 0.46		0.011
**US ratios**	**Proximal**		**Distal**	
	Mean ± SD		Mean ± SD	
D: A	0.73 ± 0.03		0.70 ± 0.03	
L: D	0.65 ± 0.01		0.67 ± 0.01	
C: D	0.34 ± 0.01		0.33 ± 0.01	
	Mean ± SD			
B: A	**0.70 ± 0.02**			
S: A	0.32 ± 0.02			

DR: digital radiology. US: ultrasonography. AN: anatomical sections. SD: standard deviation. ^a^ Column with different letter indicates non-statistical significance (*p* ≥ 0.05) using the Wilcoxon test. Bold ratios were obtained of at least one of these variables.

**Table 4 animals-13-02267-t004:** Measurements (mm) obtained from pelvic limbs hooves of alive horses through digital radiography and ultrasonography.

Variables	DR	US		*n* = 16
	Median ± SD	Median ± SD		*p* Value
Distance A	13.3 ± 1.91	10.75 ± 1.31		0.001
Tubular hoof wall (P1)	8.4 ± 2.10	6.55 ± 0.96		-
Parietal dermis (D1)	5.75 ± 1.28	7.95 ± 1.21		0.0004
Lamellar layer (L1)	-	5.4 ± 1.10		-
Sublamellar dermis (C1)	-	2.5 ± 0.26		-
Tubular hoof wall (P2)	8.35 ± 1.34	6.2 ± 0.89		-
Parietal dermis (D2)	5.15 ± 1.20	7.25 ± 1.10		0.004
Lamellar layer (L2)	-	4.9 ± 0.88		-
Sublamellar dermis (C2)	-	2.3 ± 0.36		-
Distance B	14.1 ± 2.31 ^a^	13.2 ± 0.72 ^a^		0.755
Sole dermis (S)	6.35 ± 1.47	3.4 ± 1.48		0.004
**US ratios**	**Proximal**		**Distal**	
	Mean ± SD		Mean ± SD	
D: A	0.73 ± 0.04		0.69 ± 0.04	
L: D	0.67 ± 0.01		0.67 ± 0.01	
C: D	0.32 ± 0.01		0.32 ± 0.01	
	Mean ± SD			
B: A	**0.83 ± 0.03**			
S: A	0.30 ± 0.07			

DR: digital radiology. US: ultrasonography. AN: anatomical sections. SD: standard deviation. ^a^ Column with different letter indicates non-statistical significance (*p* ≥ 0.05) using the Wilcoxon test. Bold ratios were obtained of at least one of these variables.

## Data Availability

Additional data can be found from the public accessible repository of the Universidad de Antioquia at URL (accessed on 15 December 2022) https://hdl.handle.net/10495/32382.

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
