# Peer review of "Transmural Ultrasonography in the Evaluation of Horse Hoof Internal Structures: Comparative Quantitative Findings—Part 2"

_animals, 2023, doi:10.3390/ani13142267_

Round 1

Reviewer 1 Report

The authors go deeper into the topic of ultrasound of the hoof wall to diagnose and monitor laminitis. Many variables are measured and many of them are highly dependent by by other factors. This makes the results questionable as the authors themselves demonstrated in the discussions. I think that the main limit of the paper is the language. Although I am not an English speaker I found it very difficult to understand. So I would suggest an Eden sting review of language for technical errors as well as for style.

moreover i just would like to highlight some mistakes.

line 24> something is missing in the sentence

line 30/34> please more clear in reporting the results.

line 39> just say transmural ultrasound

line 47 A proper preparation 

line 59 objective

line 78/79 figure between brackets

line 81 composed

line 87/89 i do not understand the meaning

line 121 you don’t need to specify what median is 

Line 134 a dot is present that shouldn’t be there un less something is missing in the sentence.

Line 11/122 please clarify what test, either Friedman of Wilcoxon, is used for

line 136 in which way they coincide, please clarify

line 202 i do not understand the meaning

line 210 could influence what

Reviewer 2 Report

This work has several high points. A very meticulous anatomical work, a very direct methodology. It is also easy to follow.

However, several points must be clarified for approval.

The objective is confusing... if the authors really want to validate the technique, this is not the most appropriate methodology.

The statistics must be thoroughly reviewed.

Simple Summary

the summary is very scattered. It is recommended to focus and try to describe with simplicity the work done

M&M

Very good graphic description of the determinations made with all the techniques. It is extremely easy to understand the crossed design in terms of determinations, even for a professional not used to these techniques.

The design in terms of inclusion animals and groups is not clear. Although the animals are described, the number in each technique and the type of limb (thoracic and pelvic) subjected to the different techniques is not clear. The tables in results also have no n

because the authors assume that the underlying distribution does not fit the so-called parametric criteria. The distribution must be studied, and the statistical analysis must be the most powerful (in terms of alpha and beta). this point is important.

Rx and US studies should be described in M&M. Example in x-rays shooting distance, and chassis distancei, and in US preparation characteristics 

the ratios are not described. neither how they are obtained nor what was done with them

Results

L132 these were only compared between ultrasonography and anatomical sections. 

This is M&M or can be discussed, but is not a result

L133 The other measurements were compared in the three methods. Measurements D, B, and S obtained with radiographs, ultrasonography, and anatomical sections did not show statistical significance; that is, they coincide in at least two of the analyzed methods.

Is the interpretation correct?

Discussion

Well exposed the limitations of freezing

For validation, a direct comparison was chosen... with its advantages and disadvantages. but other forms of validation are possible, which allow a broader approach. And maybe it's interesting to talk about it and discuss some concepts like repeatability and statistical difference and what they mean.

The discussion interestingly addresses the differences found from a physical point of view of light diffraction. Some of these physical concepts related to the methodology used were missed in the introduction.

the discussion is brief, and does not finish rounding off the idea of the present comparison. Many results are not discussed.

Conclusion

L290 affected in chronic laminitis. It is not supported by the results, I mean that there is no laminitis group

Round 2

Reviewer 1 Report

The authors replied to every changes I asked or suggested. I am very satisfied by the new version of the paper, which appears much more clear and readable.

I just have to highlight two spelling mistakes:

line 150 and 151: "determined".

I consider now the paper suitable for publication.

Thank you
